# Long-Term Survival and Immune Response Dynamics in Melanoma Patients Undergoing TAPCells-Based Vaccination Therapy

**DOI:** 10.3390/vaccines12040357

**Published:** 2024-03-27

**Authors:** Andrés Tittarelli, Cristian Pereda, María A. Gleisner, Mercedes N. López, Iván Flores, Fabián Tempio, Alvaro Lladser, Adnane Achour, Fermín E. González, Claudia Durán-Aniotz, Juan P. Miranda, Milton Larrondo, Flavio Salazar-Onfray

**Affiliations:** 1Programa Institucional de Fomento a la Investigación, Desarrollo e Innovación, Universidad Tecnológica Metropolitana, Santiago 8940577, Chile; atittarelli@utem.cl; 2Disciplinary Program of Immunology, Institute of Biomedical Sciences, Faculty of Medicine, Universidad de Chile, Santiago 8380453, Chile; cperda@gmail.com (C.P.); maria.gleisner@uchile.cl (M.A.G.); melopez@uchile.cl (M.N.L.); ivn.flores@gmail.com (I.F.); fabian.tempio@uchile.cl (F.T.); 3Millennium Institute on Immunology and Immunotherapy, Faculty of Medicine, Universidad de Chile, Santiago 8380453, Chile; 4Centro Científico y Tecnológico de Excelencia Ciencia & Vida, Fundación Ciencia & Vida, Santiago 8580702, Chile; alladser@cienciavida.org; 5Facultad de Medicina y Ciencia, Universidad San Sebastián, Santiago 8580702, Chile; 6Science for Life Laboratory, Department of Medicine Solna, Karolinska Institute, 17176 Stockholm, Sweden; adnane.achour@ki.se; 7Division of Infectious Diseases, Karolinska University Hospital, 17176 Stockholm, Sweden; 8Laboratory of Experimental Immunology & Cancer, Faculty of Dentistry, Universidad de Chile, Santiago 8380000, Chile; fgonzalez@uchile.cl; 9Latin American Brain Health Institute (BrainLat), Center for Social and Cognitive Neuroscience (CSCN), School of Psychology, Universidad Adolfo Ibañez, Santiago 7941169, Chile; claudia.duran@uai.cl; 10Instituto Nacional del Cáncer, Santiago 8380000, Chile; juanpablomirandaolivares@gmail.com; 11Banco de Sangre, Hospital Clínico de la Universidad de Chile, Santiago 8380453, Chile; mlarrondo@hcuch.cl

**Keywords:** immunotherapy, vaccine, melanoma, dendritic cell, clinical trials, compassionate use, overall survival, DTH

## Abstract

Cancer vaccines present a promising avenue for treating immune checkpoint blockers (ICBs)-refractory patients, fostering immune responses to modulate the tumor microenvironment. We revisit a phase I/II trial using Tumor Antigen-Presenting Cells (TAPCells) (NCT06152367), an autologous antigen-presenting cell vaccine loaded with heat-shocked allogeneic melanoma cell lysates. Initial findings showcased TAPCells inducing lysate-specific delayed-type hypersensitivity (DTH) reactions, correlating with prolonged survival. Here, we extend our analysis over 15 years, categorizing patients into short-term (<36 months) and long-term (≥36 months) survivors, exploring novel associations between clinical outcomes and demographic, genetic, and immunologic parameters. Notably, DTH^pos^ patients exhibit a 53.1% three-year survival compared to 16.1% in DTH^neg^ patients. Extended remissions are observed in long-term survivors, particularly DTH^pos^/M1c^neg^ patients. Younger age, stage III disease, and moderate immune events also benefit short-term survivors. Immunomarkers like increased C-type lectin domain family 2 member D on CD4^+^ T cells and elevated interleukin-17A were detected in long-term survivors. In contrast, toll-like receptor-4 D229G polymorphism and reduced CD32 on B cells are associated with reduced survival. TAPCells achieved stable long remissions in 35.2% of patients, especially M1c^neg^/DTH^pos^ cases. Conclusions: Our study underscores the potential of vaccine-induced immune responses in melanoma, emphasizing the identification of emerging biological markers and clinical parameters for predicting long-term remission.

## 1. Introduction

One major factor contributing to the initial resistance to immune checkpoint blockers (ICBs) in melanoma and other solid tumors is the limited infiltration of T cells into tumors (‘cold tumors’) [1]. Conversely, tumors with high lymphocyte infiltration and interferon (IFN)-γ status associated with a T cell inflamed phenotype (‘hot tumors’) exhibit significantly higher susceptibility to anti-PD-1/PD-L1 (programmed cell death protein-1 and its ligand, respectively) therapies [2]. Consequently, there is a growing interest in developing and re-evaluating active immunological treatments that induce adaptive cellular responses in cancer patients, particularly those with cold tumors, aiming to convert these into hot tumors. In this context, therapeutic cancer vaccines have emerged as an attractive alternative or complement to cancer treatment [3].

Dendritic cells (DCs) are highly efficient antigen-presenting cells responsible for orchestrating T cell activation, connecting innate and adaptive immunity [4]. The emergence of tumor-associated antigens (TAAs) has paved the way for developing TAA-pulsed autologous DCs generated ex vivo by culturing monocytes in the presence of specific cytokine combinations. This approach has been harnessed for therapeutic cancer vaccination [5]. While clinical outcomes have occasionally yielded ambiguous results, DC vaccines remain a focal point of extensive investigation, predominantly owing to their established safety profile and minimal toxicity [4,6]. Notably, the groundbreaking success story in this arena is exemplified by Sipuleucel-T, which stands as the pioneering FDA-approved cell-based therapy for patients with hormone-refractory prostate cancer [7]. Furthermore, DC-based vaccines exhibited great promise in treating various types of solid tumors.

There are numerous therapeutic cancer vaccines, including those that utilize pre-defined and known TAA or neoantigens and those that use whole tumor cells (WTC) as a source of tumor antigens and endogenous adjuvants [8]. WTC vaccines provide clinical benefits, including low risks of adverse effects, increased anti-tumor response, and, in some cases, prolonged patient survival [8]. Firstly, WTC vaccines encompass a broad range of tumor antigens, including shared and unique epitopes, enabling more comprehensive anti-tumor immune responses. This approach also increases the likelihood of targeting tumor heterogeneity and reduces the risk of tumor escape due to antigen loss variants. Furthermore, WTC vaccines can simultaneously activate multiple immune system components, including T cells (CD4^+^ and CD8^+^) and B cells. By engaging two primary arms of the immune response, these vaccines can generate robust and durable immune memory, providing prolonged protection against tumor recurrence [9,10]. Last but certainly not least, WTC can be pretreated by using different stressors to produce potent endogenous adjuvants and associated danger signals, which promote adequate immune responses mediated by appropriate subsets of DCs and functionally activated effector T cells [8].

In the early 2000 decade, our group conducted a phase I/II clinical trial in patients with advanced melanoma utilizing the WTC lysate-loaded DC vaccine named TAPCells (or TRIMEL/DC) [11,12]. The TAPCells vaccination uses peripheral blood monocyte-derived antigen-presenting cells stimulated with the allogeneic heat-shock-conditioned melanoma cell lysate TRIMEL [12]. Our clinical studies demonstrated that TAPCells treatments induced detectable immunological responses in about 60% of vaccinated patients, characterized by a delayed-type hypersensitivity (DTH) response against TRIMEL one month after treatment, associated with a threefold improvement in overall survival compared to non-responder (DTH^neg^) patients [11,12].

The DTH reaction has long been a reliable indicator of T-cell-mediated immunity and frequently showed direct correlations with clinical outcomes in our and other studies [11,12,13,14,15,16]. Moreover, DTH biopsies from responder melanoma patients included a high proportion of CD4^+^CD45RO^+^ and CD8^+^CD45RO^+^ memory T cells [12], indicating that TAPCells vaccination can induce detectable cell-mediated immune responses after treatment discontinuation.

When the results of this study were previously published, they had a median follow-up time of between 33.6 and 48 months [11,12]. Notably, DTH immune response correlated with a marked reduction in the frequency of regulatory T (Treg) cells and an increase in the proportion of T cells producing IFN-γ (Th1) and interleukin (IL)-17 (Th17) in the peripheral blood of DTH^pos^ melanoma patients [11,13]. In a subsequent report, we identified additional peripheral blood cell-derived molecular markers associated with DTH responses of TAPCells-vaccinated patients by an unbiased and comprehensive gene expression approach [14].

In the present work, we provide an update on the overall survival outcomes of the 86 patients treated within the frame of our TAPCells vaccination clinical trial with a median follow-up period of 188 months (15.6 years). As numerous immunotherapy clinical trials in melanoma have suggested that patients who remain alive at the three-year landmark will likely experience prolonged cancer remission [17], we herein categorized TAPCells-treated patients into short- (<36 months) and long-term (≥36 months) survivors and evaluated the association of their clinical responses to different demographic, medical, genetic, and immunological variables. Moreover, we also included data from 24 additional metastatic melanoma (MM) patients treated with TAPCells within a compassionate-use setting, which results have not been previously reported. This other patient cohort was followed for a similar period (median follow-up time of 155 months), demonstrating similar efficacy of TAPCells vaccination as that of patients included in the clinical trial. The TAPCells vaccine was associated with prolonged overall survival (≥36 months) in approximately 52% of immunologically responding (DTH^pos^) patients, including both cohorts.

In addition, data from patients collected at different times using flow cytometry [13,14], as well as transcriptome profiling by microarray and quantitative real-time reverse transcriptase (qRT)-PCR [14], were analyzed using the new categorization of short- versus long-term survival. These analyses allowed the identification of specific immunological phenotypes in MM patients’ peripheral blood leukocytes post-vaccination that were associated with the survival benefits of TAPCells treatment. Altogether, our results indicate that TAPCells immunotherapy induces long-term survival in a critical number of patients and unveils the emergence of immunological markers and clinical parameters helpful in predicting stable remissions in vaccinated patients.

## 2. Materials and Methods

Clinical trials cohort: Appendix A presents data for the 86 patients included in a Phase I/II clinical trial (NCT06152367), monitored from January 2001 to December 2007 (median follow-up time = 48 months; range 33 to 64 months) and from September 2006 to July 2010 (median follow-up time = 33.6 months; range 13 to 47 months) [11,12]. The eligibility criteria for vaccination have been previously described [11,12,18]. Briefly, patients were over 18 years old with histologically confirmed malignant melanoma disease with objectively measurable lymph nodes or distant metastases excluding the brain. The patient’s life expectancy was over three months, and Karnofsky’s performance was above 70% [11,12,18]. Stratification of patients was performed according to the American Joint Committee on Cancer (AJCC, version 6.0). The present study presents updated overall survival analyses of this cohort of patients, with a median follow-up period of 188 months (range = 165 to 269 months; from January 2001 to December 2022).

Compassionate-use cohort: Twenty-four (24) advanced melanoma patients (21 stage IV and 3 stage III) were treated with TAPCells vaccinations in a compassionate-use setting at the Universidad de Chile Clinical Hospital between July 2009 and March 2011 and followed until December 2022 (Appendix A). Compassionate use refers to a treatment option that allows the administration of an unapproved medication, under strict conditions, to groups of patients affected by a disease for which there are no satisfactory authorized therapies available and who are unable to participate in a clinical trial. The setting included subjects of any sex and age diagnosed with melanoma in advanced stages. Patients previously treated with immune modulator drugs, chemotherapy, or biological therapies were also recruited. Subjects with allergies to vaccine components, pregnancy or breastfeeding, brain metastasis, or evident mental incapacity to understand the information or deliver the consent were excluded.

Melanoma cell lines and cell lysate preparation: The melanoma cell lysate used in the study was named TRIMEL and derived from a mixture of three allogeneic melanoma cell lines, MEL1, MEL2, and MEL3, established from samples of metastatic lymph nodes at the Institute of Biomedical Sciences, Universidad de Chile. Cell lines constituting the TRIMEL stained positive for several TAAs [11,12]. Tumor cells from each cell line were heat-shocked at 42 °C for one hour and then incubated for two hours at 37 °C. Then, the cells were mixed in equal amounts and lysed through repeated freeze–thaw cycles in liquid nitrogen. The cell lysate was thereafter sonicated and finally irradiated with a 60 Gy dose. The protein concentration of the lysate was estimated by Bradford’s method using a Biophotometer (Eppendorf, Hamburg, Germany).

Generation of TAPCells: Leukapheresis of melanoma patients was performed in the Blood Bank Service of the University of Chile Clinical Hospital as previously described [11,12]. Leukocytes were isolated from peripheral blood mononuclear cells by density gradient separation with Ficoll–Hypaque (Axis-Shield, Oslo, Norway). These cells (3 × 10^7^/well) were then incubated in serum-free AIM-V therapeutic medium (GIBCO BLR, Invitrogen, Grand Island, NY, USA) at 37 °C, 5% CO_2_ for 2 h in 6-well plates (Becton Dickinson, Hershey, PA, USA). Non-adherent cells were removed, and the remaining ones were incubated for 24 h in a serum-free AIM-V therapeutic medium in the presence of recombinant human IL-4 (rhIL-4; 500 U/mL; US Biological, Swampscott, MA, USA) and granulocyte–macrophage colony-stimulating factor (800 U/mL; Schering Plough, Brinny Co., Cork, Ireland). DCs were loaded with 100 µg/mL of TRIMEL for maturation/activation induction. After 48 h of additional incubation, DC-like cells were recovered and cryopreserved using an automatic freezing system, Cobe Spectra (Lakewood, CO, USA).

Immunization procedure: Patients were vaccinated intradermally with 1–2 × 10^7^ TAPCells mixed with aluminum hydroxide (500 µg/mL; J.T. Baker, Phillipsburg, NJ, USA) or keyhole limpet hemocyanin (KLH; 100 µg/mL; Calbiochem, San Diego, CA, USA) to a final volume of the vaccine = 1 mL. The vaccination protocol consisted of four doses injected on days 0, 10, 30, and 50. Skin DTH tests were performed as previously described [11,12]. The study was performed in agreement with the Declaration of Helsinki and approved by the Ethical Committee for Human Research of the Universidad de Chile, Faculty of Medicine 363/290102 and the Ethical Committee for Human Research of the Clinical Hospital of Universidad de Chile 227/280803. All patients were required to understand the studies and sign an informed consent letter.

Immunological characterization: Th1 (IFN-γ+) and Th17 (IL-17+) CD4+ T cell populations were previously determined by flow cytometry in peripheral blood samples of a subgroup of patients included in the clinical trials (Appendix A), as previously reported [13]. Briefly, this analysis was performed on peripheral blood mononuclear cells (PBMC) stimulated with phorbol myristate acetate (50 ng/mL) and ionomycin (1 mg/mL) in the presence of brefeldin A (1 mg/mL, eBioscience, San Diego, CA, USA) for 5 h. Cells were incubated with anti-CD4-FITC (RPA-T4) antibody (eBioscience, San Diego, CA, USA), permeabilized with permeabilization buffer (eBioscience, San Diego, CA, USA), and incubated with anti-IL-17-PE (eBio64-DEC17) and anti-IFN-γ-PerCpCy5.5 (Clone: 4S.B3) (eBioscience, San Diego, CA, USA). Cells were analyzed on a FACSort flow cytometer (Becton–Dickinson, Franklin Lakes, NJ, USA), and data were analyzed with the WinMDI 2.9 software. Serum levels of IL-17A and transforming growth factor (TGF)-β were determined by enzyme-linked immunosorbent assay as described [13]. In the present study, we analyzed the data of these experiments after categorizing the patients according to overall survival outcome (short- or long-term survivors). The toll-like receptor (TLR)-4 Asp299Gly polymorphism was previously genotyped in a subgroup of melanoma patients for both cohorts (Appendix A) using PCR-restriction fragment length polymorphism analysis [19]. Surface expression levels of CXCR4 (C-X-C motif chemokine receptor 4), CLEC2D (c-type lectin domain family two member D), and FCGR2 (low-affinity immunoglobulin γ Fc region receptor II, also known as CD32) were determined by flow cytometry on CD4^+^ T cells (CD45^+^CD3^+^CD4^+^), CD8^+^ T cells (CD3^+^CD8^+^), B cells (CD3^−^CD19^+^), natural killer (NK) cells (CD45^+^CD16^+^CD56^+^), and monocyte (CD45^+^CD14^+^) populations from the peripheral blood of healthy donors (HD) and melanoma patients (Appendix A), before and after TAPCells vaccination, as previously described [14]. Briefly, thawed PBMCs were stained with the following antibodies, APC-conjugated anti-hCD32 (clone FLI8.26; BD Pharmingen, San Diego, CA, USA) and anti-hOCIL/CLEC2d (clone 402659; R&D Systems, Minneapolis, MN, USA); FITC-conjugated anti-hCD14 (clone 61D3) and anti-hCD16 (clone eBioCB16) (eBioscience, USA); PE-conjugated anti-hCD11c (clone 3.9; eBioscience, USA); FITC/PE-conjugated anti-hCD3/CD4 (clone SK7/SK3) and anti-hCD3/CD8 (clone SK7/SK1) (BD Simultest); PE Cy7-conjugated anti-hCD184 (clone 12G5; BD Pharmingen, USA) and PerCP-conjugated anti-hCD19 (clone HIB19) and anti-hCD45 (clone HI30) (BioLegend, San Diego, CA, USA). Cells were analyzed on a FACSCanto II (Becton Dickinson, USA) flow cytometry and performed using the FlowJo software 10.4 (Tree Star Inc., San Carlos, CA, USA). Again, in the present study, we analyzed the data of these experiments after categorizing the patients according to overall survival outcome.

Statistical analysis: Analyses were performed using GraphPad Prism 10 (GraphPad Software Inc., San Diego, CA, USA). Chi-squared with Yates correction analyses were used to examine the frequency of categorical variables in different groups. Kaplan–Meier and log-rank tests were used to construct and evaluate the overall survival data. Multivariate analyses were performed using the Cox proportional hazard regression model (Breslow method) to study the effects of different variables on the overall survival of patients. Receiver operating characteristic (ROC) curves were drawn to evaluate the variable’s predictive and cut-off values. T-test was used for continuous variables after confirming normal distribution with the Shapiro–Wilk test. Differences in expression levels of immunological markers were evaluated using the Kruskal–Wallis ANOVA (multiple comparisons) test, paired *t*-test (Wilcoxon test), or unpaired *t*-test. Differences were considered statistically significant at *p* < 0.05.

## 3. Results

### 3.1. Long-Term Remissions Are Associated with the Absence of Visceral Metastases (M1c^neg^), Previous Use of Neoadjuvant Therapies, and DTH Reaction against TRIMEL in MM Patients Vaccinated with TAPCells

Eighty-six (86) melanoma patients were included in a TAPCells clinical trial (Appendix A), as previously reported [11,12]. Among them, 66 were in stage IV, 18 were in stage III, both considered advanced metastatic high-risk conditions according to the AJCC, and two patients were in stage IB. This report presents an updated and comprehensive analysis of these patients with a median follow-up period of 188 months (range = 165 to 269 months). The average age at treatment initiation was 49.8 ± 14.7 years (44.6 ± 13.6 years for stage III and 51.4 ± 14.9 years for stage IV patients, *p* = 0.153), and the sex ratio was perfectly balanced 50/50. Only 29.1% (*n* = 25) of the patients experienced moderate immune-related adverse events (irAEs), including local erythema or mild fever. The most frequent primary tumor location was in an extremity (47.7% of the patients; *n* = 41), while 24 patients (27.9%) had the primary tumor located in the trunk and 19 (22.1%) in the head and neck area. Only two patients had primary tumors with an unknown origin. Twenty-two (22) patients underwent surgery shortly before vaccination, whereas 40 received neoadjuvant treatment (radiotherapy, dacarbazine, cyclophosphamide, tamoxifen, temozolomide, IFN-α, or IL-2) before TAPCells vaccination. Only one patient had received ipimilumab before vaccination. Twenty-four (24) patients received no additional treatment besides the vaccine (Appendix A). All the patients received only one round of four doses of the TAPCells vaccine, except four patients (MT3, MT8, MT10, and MT34) who were re-vaccinated with a second round one year after the first cycle.

The overall median survival of MM patients (*n* = 84) was 16 months, with 1-year, 3-year, and 5-year estimated survival rates of 63.1%, 36.9%, and 27.4%, respectively (Figure 1A). Immunological response to TAPCells treatments was determined by DTH reactions (induration ≥ 5 mm in diameter) against TRIMEL, assessed one month after the last immunization [11,12]. DTH reactions against TRIMEL were detected in 62.2% of vaccinated patients. MM patients with a positive DTH response (DTH^pos^) had a median overall survival of 36 months, compared with 11 months for patients without a DTH response (DTH^neg^) (*p* < 0.0001; hazard ratio (HR) = 0.39; Figure 1A). The 1-year, 3-year, and 5-year estimated survival rates for DTH^pos^ MM patients were 83.7%, 53.1% and 38.8%, whereas estimations for DTH^neg^ patients were 38.7%, 16.1% and 12.9%, respectively (5-year estimated survival *p* = 0.0004; Figure 1A). The frequency of DTH positivity was directly associated with the overall survival time interval of MM patients (Figure 1A) and indirectly related to disease severity according to AJCC classification: 83.3% for patients with N disease (N2a-N3, stage III), 65% for patients with M1a disease, 58.8% for patients with M1b disease, and only 45.8% for patients with M1c disease (Figure 1B).

Multivariate analysis (Figure 1C) confirmed the observation that DTH^pos^ patients were more likely to experience longer overall survival times compared to DTH^neg^ patients (HR = 0.33; 95% confidence interval (CI), 0.19 to 0.56; *p* < 0.0001). In contrast, patients with stage IV (HR = 2.9; 95% CI, 1.35 to 6.84; *p* = 0.009) or with M1c disease (HR = 2.04; 95% CI, 1.17 to 3.54; *p* < 0.011) were less likely to have longer overall survival after TAPCells vaccination compared to those with stage III or other metastasis stages, respectively. Melanoma patients previously treated with any kind of neoadjuvant therapy (excluding surgery) showed a tendency to longer overall survival times (HR = 0.58; 95% CI, 0.33 to 1; *p* = 0.053) compared with patients who did not receive neoadjuvant therapies (Figure 1C). Eighteen (18) MM patients exhibited the co-occurrence of a positive DTH, the use of neoadjuvant therapy, and the absence of metastases in organs other than the lungs (M1c^neg^). Kaplan–Meier analysis showed that this group of patients had a more than fourfold increase in median overall survival time (57.5 months) compared with the rest of the MM patients (13 months; *p* = 0.0026; Figure 1D). The 3-year estimated survival for MM patients who exhibited the co-occurrence of the variables above was 72.2%, whereas, for the rest, it was only 26.1% (Figure 1D). Finally, ROC curve analysis suggested an excellent predictive value for these variables (area under the curve (AUC) = 0.78, *p* = 0.0002; Figure 1E).

### 3.2. Short- and Long-Term Overall Survival Is Differentially Impacted by Demographic and Clinical Variables in TAPCells-Vaccinated MM Patients

Numerous immunotherapeutic clinical trials in melanoma have suggested that patients who survive beyond the 3-year landmark will likely experience prolonged cancer remission [17]. Therefore, we categorized TAPCells-treated MM patients into two groups: short- (<36 months; median overall survival time = 11 months) and long-term survivors (≥36 months; median overall survival time = 152 months) and analyzed separately the impact of demographic and clinical factors (Figure 2; Table 1 and Appendix A). Short- and long-term survivors did not show significant differences in age at treatment initiation, sex, primary tumor location, type of adjuvant, neoadjuvant therapy used, or occurrence of irAEs (Table 1). On the other hand, the frequencies of patients with positive DTH reactions, stage III disease, or disease stage different from M1c (M1c^neg^) were higher in long-term compared with short-term survivors (*p* = 0.001; *p* = 0.033; *p* = 0.022, respectively; Table 1).

The overall survival time of short- and long-term survivor MM patients was not affected by the use of neoadjuvant therapies, the type of adjuvant, the primary tumor location, or the sex of the patients (Appendix A). In contrast, the age at treatment initiation, the occurrence of irAEs, and the DTH reaction significantly impacted the TAPCells vaccination outcome only in short-term survivors (Figure 2). Thus, patients older than the cohort’s average age (≥50 years old) showed poorer median overall survival times than younger patients (8 vs. 12 months; *p* = 0.028; Figure 2B). Short-term survivor patients with positive DTH reactions or who experienced some type of irAEs in response to TAPCells showed increased median overall survival times compared with DTH^neg^ patients or with those who did not experience irAEs (14 vs. 8.5 or 14 vs. 8.5 months; *p* = 0.003 or *p* = 0.0034, respectively; Figure 2C,F). As expected, stage III had longer survival times than stage IV patients (median overall survival time = 65 vs. 14.5 months; *p* = 0.0017; Figure 2DI). The same was observed when short-term survivors were analyzed independently (14 vs. 9.5 months; *p* = 0.0299). However, the difference between the median overall survival times of stage IV and stage III long-term survivors did not reach statistical significance (*p* = 0.0598; Figure 2DIII).

The overall survival times for the complete cohort of clinical trial patients with metastases in the lungs (M1b) or visceral metastasis (M1c) were statistically similar (*p* = 0.43). No significant differences were observed between patients with lymph nodes (N2a-N3) or in-transit skin (M1a) metastases (*p* = 0.318; Figure 2EI). The overall survival of patients with N/M1a diseases was statistically more prolonged than that of patients with M1b/M1c (*p* < 0.01; Figure 2EI). Similar differences were observed between N/M1a and M1b/M1c for long-term survivor patients (*p* = 0.0075; Figure 2EIII). However, the median overall survival for short-term survivors was statistically different between patients with N/M1a/M1b diseases and M1c disease (13 vs. 7 months, *p* = 0.01; Figure 2EII).

### 3.3. TAPCells Therapy Showed Similar Efficacy in a New Cohort of Patients Treated in a Compassionate-Use Setting

Based on the promising results of our clinical trial [11,12], we initiated in July 2009 a compassionate-use program for TAPCells in melanoma patients. These patients accomplished the same inclusion/exclusion criteria as for the TAPCells clinical trials. Here, we present the results for 24 MM patients who were part of the compassionate-use setting and had a similar follow-up time with patients included in the previously reported clinical trial (median follow-up time of 155 months; range 143 to 164 months; Appendix A). Except that the compassionate-use cohort members received only KLH as an adjuvant, this cohort did not differ from patients included in the clinical trial in any demographic or clinical characteristics (Table 2). Among the compassionate-use patients, 21 were in stage IV, and three were in stage III. Fifty percent (50%) of the patients were female, and the average age at treatment initiation was 51.2 ± 14.3 years. The percentage of DTH^pos^ patients was 62.5% (15 of 24). Furthermore, the frequency of long-term survivor patients (overall survival ≥ 36 months; undefined median survival time; Figure 3A) was statistically similar to that observed for patients included in the clinical trials: 29.2% vs. 36.9% (*p* = 0.647; Appendix A).

The overall median survival of MM patients in the compassionate-use setting was 16.5 months, nearly the same as the 16 months observed for the clinical trial cohort (Figure 3B). DTH^pos^ MM patients from the compassionate-use setting showed a median overall survival of 34 months, while only 9 months for DTH^neg^ patients (*p* = 0.0005; HR = 0.25). These findings are consistent with our previous clinical trial cohort observations (Figure 3C) and thus indicate that TAPCells therapy exhibits similar clinical efficacy in MM patients, regardless of whether they were included in clinical trials or compassionate-use settings.

After observing the similar efficacy of TAPCells in both cohorts, we evaluated the impact of TAPCells vaccination on the entire group of MM patients (clinical trial + compassionate use; *n* = 108 patients). Again, multivariate analysis (Figure 3D) confirmed that DTH^pos^ patients had a higher likelihood of experiencing longer overall survival times compared with DTH^neg^ patients (HR = 0.308; 95% CI, 0.19 to 0.49; *p* < 0.0001). Additionally, patients with stage IV (HR = 2.235; 95% CI, 1.16 to 4.61; *p* = 0.021) or with M1c (HR = 1.737; 95% CI, 1.07 to 2.78; *p* < 0.022) disease were less likely to have more prolonged overall survival after TAPCells vaccination, compared to those with stage III or other metastasis stages (Figure 3D). Fifty-one (51) MM patients had the co-occurrence of DTH^pos^ and M1c^neg^ characteristics. Kaplan–Meier analysis demonstrated that these patients had a more than threefold increase in overall survival time (40 months) compared to the rest of the MM patients (median overall survival time = 11 months; *p* < 0.0001; HR = 0.371; Figure 3E). The 3-year estimated survival for advanced DTH^pos^/M1c^neg^ melanoma patients was 54.9%, whereas it was only 17.5% for the remaining patients (Figure 3E). ROC curve analysis showed that the co-occurrence of these two variables has an excellent predictive value (AUC = 0.79, *p* < 0.0001; Figure 3F).

### 3.4. Phenotypic Signatures Associated with Long-Term Overall Survival in MM Patients Treated with TAPCells Vaccination

We have previously provided evidence of the significance of monocyte TLR4 expression in the TRIMEL-mediated ex vivo functional differentiation of TAPCells and the impact of a TLR4 hypofunctional single-nucleotide polymorphism (Asp299Gly; rs4986790) on the clinical efficacy of TAPCells in melanoma patients [19]. Here, our updated findings reaffirm a robust association between the Asp299Gly polymorphism (D/G or G/G) with poorer median overall survival times (13 months) when compared with patients carrying the wild-type TLR4 allele (D/D) (28 months; *p* = 0.048; HR = 0.498; Figure 4A). Indeed, patients with the TLR4 Asp299Gly polymorphism had almost a 50% reduction in survival odds beyond 36 months.

We also identified in previous reports that gene and protein expressions of CLEC2D, CXCR4, and FCGR2 (CD32) on peripheral blood leukocytes from TAPCells-treated MM patients correlate with vaccine-induced DTH responses [14]. Here, we analyzed these data by categorizing the patients into short- and long-term survivors (Table 1 and Appendix A; underlined patients) irrespective of their DTH responses. Our aim was to assess the association of peripheral blood biomarkers with extended remission in MM patients treated with TAPCells therapy. Our results, depicted in Figure 4B, showed that after TAPCells therapy (samples obtained one month after the last vaccine dose), there was a significant induction in the surface expression of CLEC2D on peripheral blood CD4^+^ T cells only in long-term survivors.

Moreover, CD4^+^ T cells from long-term survivor patients post-TAPCells therapy have significantly higher expression levels of cell surface CLEC2D than CD4^+^ T cells isolated from the peripheral blood of HD. Our results showed a significant increase in CXCR4 expression on peripheral blood monocytes after TAPCells therapy only in short-term survivor patients (Figure 4B). We also observed that CD32 surface expression in peripheral blood B cells was significantly lower after TAPCells therapy only in short-term survivors. Moreover, post-TAPCells therapy, B cells from short-term survivor patients had substantially lower levels of CD32 surface expression than B cells isolated from HD. In contrast, the expression levels of CD32 on monocytes and NK cells (Figure 4B) were increased after TAPCells only in short-term survivor MM patients. The results without statistical significance are shown in Appendix A.

We previously reported that DTH responses induced by TAPCells correlated with an increase in the proportion of Th1 and Th17 CD4^+^ T cell populations in the peripheral blood of DTH^pos^ patients [13]. Here, we reanalyzed these data (Appendix A; patients highlighted in red), categorizing the patients into short- and long-term survivors. The new analysis did not show a statistical association between the pre- and post-TAPCells therapy levels of peripheral blood Th1/Th17 CD4^+^ T cells with the overall survival outcomes of MM patients (Appendix A). However, only long-term survivor patients showed a significant increase in serum IL-17A levels after TAPCells therapy (Figure 4C).

## 4. Discussion

Cancer vaccines harbor the potential to stimulate de novo and enduring memory immune responses, presenting an appealing alternative or complement for solid tumor patients refractory to ICB or targeted therapies. Notably, some DC vaccines, such as the FDA-approved personalized immunotherapy against advanced prostate cancer Sipuleucel-T (Provenge), have shown efficacy, improving patient survival [7]. Conversely, melanoma vaccines have shown, in general, weak clinical effectiveness [6,20]. Remarkably, our studies demonstrate significant clinical effects of TAPCells anti-melanoma vaccine, showing 5-year overall survival rates of 25.9% in the entire cohort and a markedly higher overall survival rate of 41.2% for patients with MM with detectable immune responses (DTH^pos^) in the absence of visceral metastases (M1c^neg^). These results consider 108 patients included in clinical trials and compassionate-use settings.

Another DC vaccine for MM patients has shown clinical success, resulting in a median overall survival time of 49.4 months and a 5-year survival rate of 46% [21,22]. While these data are impressive, it is essential to note that 66.7% of the patients participating in this study had no detectable metastasis at the treatment time, a significant distinction from our cohort used in the present study. Specifically, MM patients with measurable metastasis (*n* = 24) exhibited a median overall survival of 18.5 months and a 46% 2-year overall survival rate [22]. These outcomes align more closely with those observed in our cohort.

Recently, we published a review of clinical trials assessing whole tumor cell vaccines [8], including those conducted in melanoma patients. Noteworthy, among these vaccines are: Canvaxin, a vaccine composed of allogeneic irradiated tumor cells combined with BCG as an adjuvant [15]; VigilTM (FANGTM), a vaccine that included autologous tumor cells genetically modified to express GM-CSF [23] (NCT01453361); Melacine, an allogeneic tumor cell lysate vaccine that used MLPA as adjuvant [24], currently under investigation in combination with IFN-α in a phase III clinical trial (NCT00002767); and Vaccimel, a vaccine composed of allogeneic tumor-irradiated cells combined with BCG and GM-CSF as adjuvants [25]. Remarkably, in a phase III study of postsurgical adjuvant therapy involving approximately 500 stage IV melanoma patients, Canvaxin’s efficacy was compared with a placebo. The results indicated that BCG/Canvaxin did not demonstrate improved outcomes over BCG/placebo [15].

The accumulation of long-term data from various immuno-oncology agent trials in melanoma has highlighted an intriguing trend in which the overall survival curves tend to level off after 3–4 years of treatment. This fact suggests that patients who surpass the 3-year milestone are likely to experience extended periods of cancer remission [17]. Illustrative clinical studies show that overall survival rates changed minimally between three and five years, with a difference of just six to ten percentage points [26,27,28]. Our MM cohorts exhibited a comparable difference of approximately nine percentage points between the overall survival rates at three and five years (Figure 1A and Figure 3E). These numbers support using the 3-year mark as a critical threshold for categorizing TAPCells-treated MM patients as short- or long-term survivors.

In line with our original reports, anti-tumor DTH reactions are still a favorable prognostic factor for long-term survivors, particularly in patients in stage III. In contrast, short-term survivors were more prevalent in patients with M1c disease (Table 1). These outcomes align with various studies that have consistently demonstrated that the most favorable survival outcomes tend to occur in patients harboring positive prognostic factors, particularly those characterized by a low disease burden [15]. The observed DTH reactions against tumor antigens strongly correlated with the melanoma patients’ overall survival, indicating that the vaccine-induced cellular-mediated immune response may be capable of controlling the disease progression. The DTH reaction has long been a reliable indicator of T-cell-mediated immunity in our and other studies [11,12,13,14,15,16,29,30]. Although very rare, the DTH^neg^ MM patients (*n* = 5) did not show statistically significant differences in median overall survival with the DTH^pos^ patients in the long-term survivor group (*n* = 26) (Figure 2CIII). Furthermore, although most DTH^neg^ patients did not exceed the 3-year survival threshold, a more exhaustive examination of immune parameters for these patients could reveal ideas to improve their survival outcomes.

Regarding the patient’s age at treatment initiation and the occurrence of irAEs during the treatment, these variables did not exhibit a statistically significant association with the overall survival in the entire clinical trial cohort. However, intriguing trends emerged within the subgroup of short-term survivors, where younger MM patients and those who experienced irAEs showed notably improved overall survival compared to older counterparts without irAEs (Figure 2B). Therefore, the overall survival outcomes of patients who did not benefit from TAPCells might develop an inadequate response due to the effects of immune senescence [31]. Further, mild and manageable irAEs are associated with constructive triggering of the anticancer immune response [32]. These irAEs could be due to various triggers, such as possible reactions against autoantigens, collateral damage underlying cytokine-induced inflammation, antigen-specific T cell reactions, or activation of B cell responses [33,34].

Notably, the primary outcomes obtained with the extensive longitudinal data from the reported clinical trial align harmoniously with the clinical results observed in a relatively small (*n* = 24) yet representative cohort of MM patients who participated in a compassionate-use program (Figure 3). These patients constituted the inaugural beneficiaries of this compassionate-use initiative, benefitting from a follow-up duration akin to that of the clinical trial cohorts.

Different analyses suggest that early changes in peripheral blood markers after vaccination may be associated with survival outcomes in treated patients. Our study observed increased CXCR4 expression in monocytes following TAPCells therapy, primarily in short-term survivor patients. The G-protein-coupled chemokine receptor CXCR4 expression in monocytes has been linked to pro-tumor effects, especially when co-cultured with tumor cells like ovarian cancer and melanoma [35,36]. These CXCR4-expressing monocytes exhibit characteristics of myeloid-derived suppressor cells, capable of suppressing T cell proliferation and IFN-γ production [36]. Preclinical research across various cancers demonstrated that CXCR4^+^ monocytes recruited by tumor cells promote tumor angiogenesis [37,38,39]. Inhibiting CXCR4-dependent monocyte recruitment enhances the effectiveness of antiangiogenic therapy and anti-PD-1 immunotherapy [40,41]. CXCR4 antagonists also reduced melanoma lung metastasis in mice [42]. Notably, CRC patients with poorer prognoses had higher CXCR4 expression in peripheral blood CD14^+^ monocytes than patients with better prognoses [43].

Further analyses showed that CLEC2D expression increases in peripheral blood CD4^+^ T cells in long-term survivor melanoma patients after vaccination. CLEC2D, or lectin-like transcript 1 (LLT1), acts as the ligand for the receptor CD161 present in both NK and T cells [44]. Numerous activating signals can trigger the expression of CLEC2D, including T cell receptor cross-linking and cytokine stimulation [45]. Interestingly, CLEC2D high-density expression on tumor-infiltrating lymphocytes (TILs) in oropharyngeal squamous cell carcinoma (OPSCC) correlates with the highest survival rates [46,47]. Given that CLEC2D induction requires robust and sustained immune activation [48], it would be interesting to investigate the potential of CD4^+^LLT1^+^ T cells as peripheral blood predictive biomarkers for immunotherapy responses and cancer remission.

The levels of low-affinity receptor FcγRII (CD32) serve as an indicator for the B-cell activation state. Interestingly, our analyses post-vaccination showed that peripheral blood B cells from short-term survivor patients have decreased surface levels of CD32 compared with B cells isolated from HDs or the pre-vaccination blood samples. These observations suggest that B cells with low expression of FcγRII in peripheral blood may have a relevant role as a cellular predictor of poor long-term outcomes in patients vaccinated with TAPCells (AUC = 0.826, 95% CI = 0.65–0.99, *p* = 0.0095; ROC analysis).

A significant accumulation of activated B cells with low expression of (FcγRII^low/neg^) at the invasive margin has been observed in hepatocellular carcinoma tumors [49]. These FcγRII^low/neg^ B cells exhibited a noteworthy ability to spontaneously produce IL-10 ex vivo that could suppress the expression of cytotoxic granzyme B and perforin, as well as the pro-inflammatory cytokines such as tumor necrosis factor-α and IFN-γ, in autologous tumor-derived CD8+ cytotoxic T cells [49].

CD32 is a receptor with two activating isoforms, CD32A and CD32C, while CD32B is inhibitory [50]. NK cells express mainly CD32C and less prevalently CD32B in NK cells, whereas monocytes mainly express the A and B isoforms [51,52,53,54,55]. Notably, FcγRIII (CD16)-mediated antibody-dependent cellular cytotoxicity (ADCC) by NK cells is enhanced by monocytes but suppressed by CD32 cross-linking using anti-CD32 monoclonal antibodies (mAb, clone FL18.26) [56]. Our cohort’s analysis of peripheral blood shows increased CD32 expression in monocytes and NK cells after TAPCells vaccination, particularly in patients with shorter survival, which may suggest augmented ADCC-regulating cells in such patients. Intriguingly, ovarian cancer patients receiving other autologous DC vaccines with strong vaccine-induced ADCC responses had lower recurrence rates [57]. While we did not measure our vaccine-induced anti-melanoma humoral responses within the present study, the TRIMELVax vaccine that contains TRIMEL lysates induced anti-melanoma humoral responses in mice [58].

Studies evaluating the association between IL-17 and patients’ prognoses are inconsistent across various cancer types [59]. However, a recent study has shown that high levels of IL-17A in the baseline peripheral blood of patients with melanoma are associated with better clinical responses to dual ICB therapy (anti-PD-1 + anti-CTLA-4) [60]. In line with these observations, our study showed that the post-TAPCells fold change in IL-17A plasma levels in long-term survivor MM patients was significantly higher than in short-term survivors. However, the limited number of samples analyzed makes it necessary to confirm this finding in more patients.

In addition, our results validate that germ-line genetic differences in TLR4 can predict long-term overall survival in MM patients treated with TAPCells. We previously demonstrated that TLR4 plays an essential role in TRIMEL-mediated phenotypic and functional differentiation of monocytes into mature DC-like cells. DAMPs in TRIMEL increase DC surface marker expression, cytokine gene expression, and the capacity to activate melanoma-specific TILs via TLR4 [19]. The negative association between the TLR4 Gly299 polymorphic allele and a poorer overall survival prognosis has also been reported in head and neck squamous cell carcinoma treated with systemic adjuvant therapies [61] and colon cancer treated with chemotherapy [62].

## 5. Conclusions

A comprehensive retrospective analysis of a long-term patient cohort demonstrates that TAPCells-mediated induction of a detectable immune response correlates with extensive and durable remissions in several patients, even without supplementary interventions. The current study focuses on a patient group vaccinated between 2003 and 2008, during which ICB therapies and other forms of immunotherapy had yet to receive regulatory approval. In Chile, the public health coverage for using ipilimumab and nivolumab in treating metastatic melanoma was only initiated in 2019.

Our findings strongly indicate that the TAPCells vaccine can elicit enduring overall survival benefits in a substantial cohort of MM patients. The remarkable extended survival observed among numerous metastatic patients underscores the possibility that an effective immune response elicited through vaccination, particularly under favorable conditions such as low tumor burden, may suffice to halt the progression of the disease. Correlations found between the duration of remission in patients and cellular and molecular markers in their peripheral blood allow progress in the search for early markers of efficacy that will undoubtedly enable a more precise selection of patients and increase the response rate of cancer vaccines.

## Figures and Tables

**Figure 1 vaccines-12-00357-f001:**
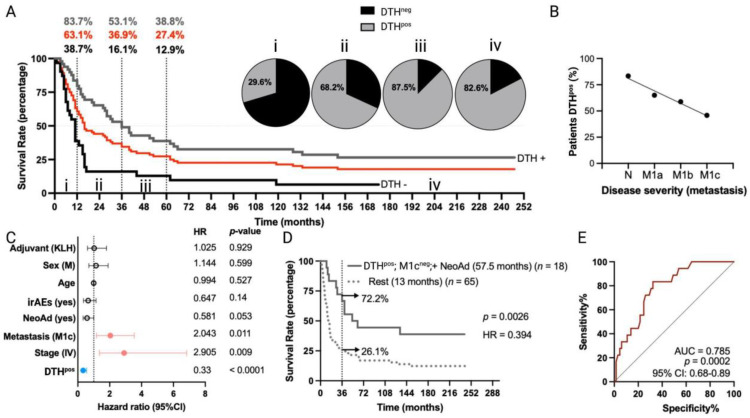
Long-term overall survival times of MM patients included in TAPCells clinical trials are associated with the lysate-specific DTH response, the use of neoadjuvant therapy, and the absence of visceral metastases. (**A**) Kaplan–Meier overall survival estimation post-TAPCells treatment for the complete cohort of metastatic (stages III and IV) melanoma patients (*n* = 84 patients; red line; median survival time = 16 months). Gray and black lines show the survival curves for patients DTH^pos^ (*n* = 49; median survival time = 36 months) and DTH^neg^ (*n* = 31; median survival time = 11 months). The percentage of patients (gray: DTH^pos^; black: DTH^neg^; red: complete cohort) alive at 12 months, 36 months, and 60 months post–therapy and the frequency of DTH^pos^ patients in each survival time range are shown (i–iv). (**B**) Percentage of DTH^pos^ patients according to their disease severity. (**C**) Cox multiple regression analysis for MM patients treated with TAPCells (*n* = 84). The figure shows the actual hazard ratios (HR) and *p*-values on the right. Statistically significant HR are highlighted in colors (light red: higher hazard; light blue: lower hazard). (**D**) Kaplan–Meier post-TAPCells treatment overall survival estimation for MM patients according to the co-occurrence of positive prognosis factors (DTH^pos^/M1c^neg^/with NeoAd). The median overall survival times in months, the number of patients included in each group (*n*), the *p*-value, the associated HR, and the percentage of long-term survivors (≥36 months) are shown. (**E**) Receiver operating characteristic (ROC) curve for MM patient overall survival according to the co-occurrence of the positive prognosis factors. irAEs: immune-related adverse events; AUC: area under the curve; CI: confidence interval; DTH: delayed-type-IV hypersensitivity; irAES: immune-related adverse events; KLH: keyhole limpet hemocyanin; M: male; NeoAd: neoadjuvant treatments.

**Figure 2 vaccines-12-00357-f002:**
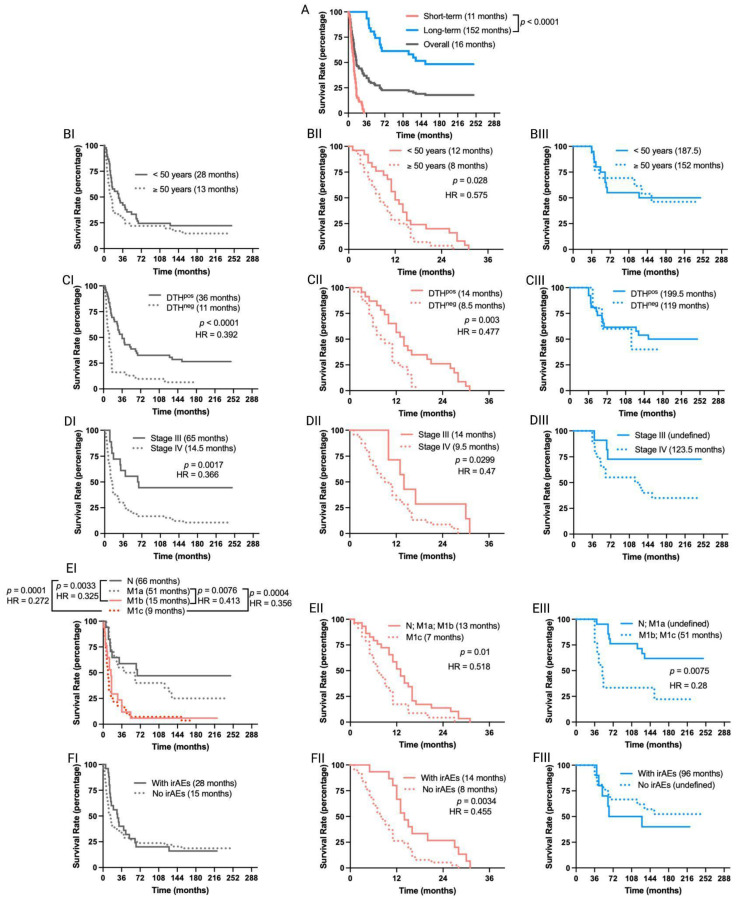
Overall survival outcomes are differentially affected in short-term and long-term survivors of MM patients included in TAPCells clinical trials. (**A**) Kaplan–Meier post-vaccine treatment overall survival estimation for MM patients according to their categorization in short- (<36 months; *n* = 53 patients) or long-term (≥36 months; *n* = 31 patients) survivors. The gray line shows the overall survival for all patients. The median overall survival times in months and the *p*-value are shown. (**B**–**F**) Kaplan–Meier post-vaccine treatment overall survival estimation for MM patients (**I**: complete cohort; **II**: short-term survivors; and **III**: long-term survivors) according to age at treatment initiation (**B**), DTH response (**C**), tumor stage (**D**), metastasis (**E**), and occurrence or not of immune-related adverse events (irAEs; (**F**)). The median overall survival times in months, the *p*-value, and the associated hazard ratios (HR) are shown only when statistically significant differences were observed. N: with metastases in lymph nodes; M1a: in-transit skin metastases; M1b: lung metastasis; M1c: metastases in organs other than lungs. DTH: delayed-type-IV hypersensitivity.

**Figure 3 vaccines-12-00357-f003:**
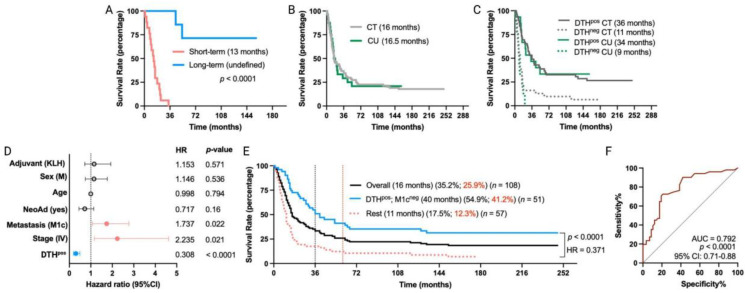
Long-term remission in MM patients treated with TAPCells in a compassionate-use setting. (**A**) Kaplan–Meier post-vaccine treatment overall survival estimation for MM patients, including in the compassionate-use (CU) setting according to their categorization in short-term (*n* = 17 patients) or long-term (*n* = 7 patients) survivors. The median overall survival times in months and the *p*-value are shown. (**B**,**C**) Kaplan–Meier post-vaccine treatment overall survival estimation for MM patients according to their clinical setting and DTH response. (**D**) Cox multiple regression analysis for all MM patients treated with TAPCells, included in the clinical trials and the CU setting (*n* = 108 patients). The figure shows the actual hazard ratios (HR) and *p*-values on the right. Statistically significant HR are highlighted in colors (light red: higher hazard; light blue: lower hazard). (**E**) Kaplan–Meier post-vaccine treatment overall survival estimation for all MM patients (including in both clinical trials + CU setting) according to the co-occurrence of positive prognosis factors (DTH^pos^/M1c^neg^). The median overall survival times in months, the 3-year (black) and 5-year (red) survival rates, the number of patients included in each group, the *p*-value, and the associated HR are shown. (**F**) Receiver operating characteristic (ROC) curve for all the MM patient survival (included in clinical trial + CU settings) according to the co-occurrence of the favorable prognosis factors. AUC: area under the curve; CI: confidence interval; CT: clinical trial; DTH: delayed-type-IV hypersensitivity; KLH: keyhole limpet hemocyanin; M: male.

**Figure 4 vaccines-12-00357-f004:**
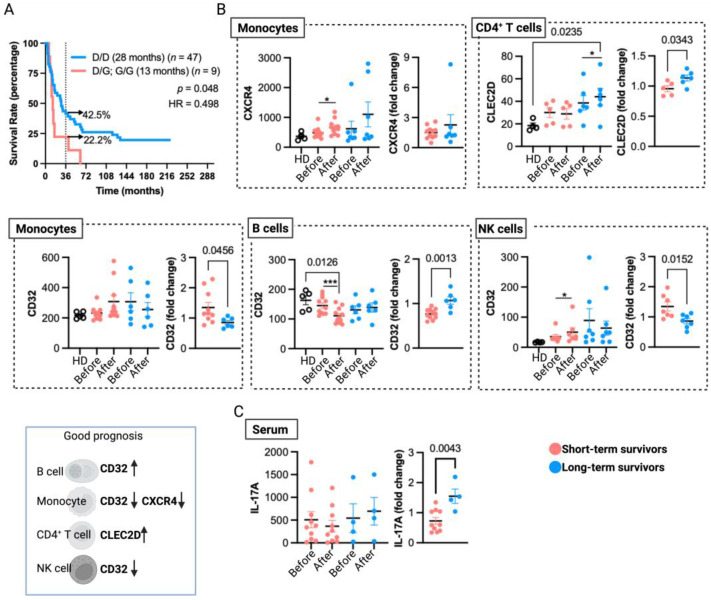
Immunological aspects associated with long-term overall survival in MM patients treated with TAPCells. (**A**) Kaplan–Meier post-TAPCells treatment overall survival estimation of MM patients (included in clinical trials and compassionate use) according to their TLR4 genotype. The median overall survival times in months, the number of patients included in each group, the *p*-value, the associated hazard ratio (HR), and the percentage of long-term survivors (≥36 months) are shown. (**B**) Cryopreserved peripheral blood mononuclear cells from healthy donors (HD; *n* = 4–5), short- (*n* = 5–12), and long-term (*n* = 5–9) survivor patients, before and after TAPCells immunization protocol, were analyzed for CLEC2D, CXCR4, and CD32 surface expression in monocytes, CD4^+^ T cells, monocytes, and natural killer (NK) cells. Each data point represents one patient sample. The graphs on the left figures show the mean fluorescence intensity of each marker in the corresponding cell type; the graphs on the right show the post-therapy fold change of each marker. A schematic summary of cell marker expressions (indicated by arrows) associated with a good prognosis for TAPCells vaccination is shown at the bottom left. (**C**) The serum levels of IL-17A (pg/mL) for short- (*n* = 10) and long-term (*n* = 4) survivors at the beginning (pre-TAPCells) and at the end (post-TAPCells) of immunization protocol were analyzed by ELISA. Statistically significant *p*-values are shown. * *p* < 0.05; *** *p* < 0.001, paired *t*-test (Wilcoxon test).

**Table 1 vaccines-12-00357-t001:** Demographic characteristics of short-term and long-term survivors of TAPCells-vaccinated melanoma patients included in clinical trials.

Variable	Short-Term Survivors	Long-Term Survivors	*p*-Value
**DTH**			(*)
Positive	23	28	
Negative	26	5	**0.001**
**Sex**			(*)
Female	24	19	
Male	29	14	0.375
**Age** (mean ± SD)	51.6 ± 15.1	46.8 ± 13.9	0.14 (#)
**AJCC criteria**			(*)
Stage III	7	11	
Stage IV	46	20	**0.033**
**Metastasis**			(*)
M1a	9	11	0.089
M1b	13	4	0.338
M1c	23	5	**0.022**
N	7	10	0.063
**Primary tumor**			(*)
Head and neck	10	9	0.518
Trunk	18	6	0.18
Limb	25	16	0.918
Unknown	0	2	0.281
**Neoadjuvant treatment (**)**			(*)
Yes	20	18	
None	33	15	0.192
**Adjuvant**			(*)
KLH	37	19	0.355
Al(OH)_3_	6	6	0.567
KLH + Al(OH)_3_	2	3	0.582
KLH + IL-2	3	3	0.863
Al(OH)_3_ + IL-2	5	1	0.485
KLH + Al(OH)_3_ + IL-2	0	1	0.81
**irAEs**			(*)
Yes	15	10	
None	38	23	0.964

AJCC: American Joint Committee on Cancer; DTH: delayed-type hypersensitivity against TRIMEL; irAEs: immune-related adverse events; KLH: keyhole limpet hemocyanin; SD: standard deviation. (*) Chi-square test (Yale’s correction). N: with metastases in lymph nodes; M1a: in-transit skin metastases; M1b: lung metastasis; M1c: metastases in organs other than lungs. (#) Unpaired *t*-test. (**) Surgery was not considered a neoadjuvant treatment.

**Table 2 vaccines-12-00357-t002:** Comparison of demographic characteristics between clinical trial and compassionate-use TAPCells-treated melanoma patient cohorts.

Variable	CT Patients	CU Patients	*p*-Value
**DTH**			(*)
Positive	51	15	
Negative	31	9	0.832
**Sex**			(*)
Female	43	12	
Male	43	12	0.817
**Age** (mean ± SD)	49.8 ± 14.7	51.2 ± 14.3	0.662 (#)
**AJCC criteria**			(*)
Stage III	18	3	
Stage IV	66	21	0.495
**Metastasis**			(*)
M1a	20	5	0.93
M1b	17	10	0.071
M1c	28	6	0.551
N	17	3	0.542
**Primary tumor**			(*)
Head and neck	19	3	0.453
Trunk	24	4	0.394
Limb	41	14	0.489
Unknown	2	3	0.118
**Neoadjuvant treatment (**)**			(*)
Yes	38	12	
None	48	12	0.784
**Adjuvant**			(*)
Without KLH	18	0	
With KLH	68	24	0.032

AJCC: American Joint Committee on Cancer; CT: clinical trial; CU: compassionate use; DTH: delayed-type hypersensitivity against TRIMEL; KLH: keyhole limpet hemocyanin; SD: standard deviation. N: with metastases in lymph nodes; M1a: in-transit skin metastases; M1b: lung metastasis; M1c: metastases in organs other than lungs. (*) Chi-square test (Yale’s correction). (#) Unpaired *t*-test. (**) Surgery was not considered a neoadjuvant treatment.

## Data Availability

All published data and material are available under reasonable request.

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
