# Peer review of "Long-Term Survival and Immune Response Dynamics in Melanoma Patients Undergoing TAPCells-Based Vaccination Therapy"

_vaccines, 2024, doi:10.3390/vaccines12040357_

Round 1
Reviewer 1 Report
Comments and Suggestions for Authors
Congratulations to the authors for this very detailed and interesting work.
Could the authors elaborate on BRAF and other molecular markers in your patient population, and how these results may potentially impact your findings?
Author Response
Answer: We thank the reviewer for the positive remarks and appreciate the insightful inquiry into BRAF and other molecular markers in our patient population. The impact of these markers on our findings is a crucial aspect of our study.
BRAF, a key player in melanoma pathogenesis, exhibits mutations in approximately 50% of melanomas, significantly influencing therapeutic responses. Notably, these mutations can have profound implications for the sensitivity of tumors to immunotherapy, directly correlating with observed treatment responses. While our clinical study unfortunately faced limitations in access to primary tumors and metastases, the acknowledged significance of BRAF and other molecular markers in melanoma warrants further exploration.
Notably, BRAF inhibitor therapies, a pivotal component in melanoma treatment, received regulatory approval for the first time in 2011, postdating the period covered by the clinical studies analyzed in our work. Consequently, the absence of data on BRAF inhibitors in our study is attributed to the temporal context, and future investigations incorporating these therapies could provide valuable insights into their impact on therapy outcomes.
Reviewer 2 Report
Comments and Suggestions for Authors
The authors present an interesting update of their phase I/II clinical trial started in early 2000, using TAPCells vaccine conducted on patients with advanced melanoma.
For this analysis, the follow-up time is 15,6 years. Moreover, 24 metastatic melanoma patients who were part of the compassionate use setting were included.
The description of the patient's cohorts is very detailed and exhaustive.
Below you can find my comments on the rest of the paper:
- Please better explain the definition of “compassionate use cohort” compared to the previous clinical trial cohort.
- Please include in the introduction or the discussion more data on other vaccines/clinical trials on melanoma patients
- Figure 4 has to be reformatted. The presentation of the different plots is very confusing and difficult to read. Moreover, please add more details on the flow cytometry procedure in the material and method section.
Author Response
- Please better explain the definition of “compassionate use cohort” compared to the previous clinical trial cohort.
Answer: We sincerely thank the Reviewer’s comments and incorporated suggested clarifications.
Compassionate use refers to a treatment option that allows the administration of an unapproved medication, under strict conditions, to groups of patients affected by a disease for which there are no satisfactory authorized therapies available and who are unable to participate in a clinical trial.
(https://www.ema.europa.eu/en/human-regulatory-overview/research-and-development/compassionate-use#:~:text=Compassionate%20use%20is%20a%20treatment,who%20cannot%20enter%20clinical%20trials).
- Please include in the introduction or the discussion more data on other vaccines/clinical trials on melanoma patients
Answer: We have included more data about clinical trials for melanoma vaccines in the discussion section: “Recently, we published a review of clinical trials assessing whole tumor cell vaccines [8], including those conducted in melanoma patients. Noteworthy, among these vaccines are: Canvaxin, a vaccine composed of allogeneic irradiated tumor cells combined with BCG as an adjuvant [15]; VigilTM(FANGTM), a vaccine that included autologous tumor cells genetically modified to express GM-CSF [60] (NCT01453361); Melacine, an allogeneic tumor cell lysate vaccine that used MLPA as adjuvant [61], currently under investigation in combination with IFN-a in phase III clinical trial (NCT00002767); and Vaccimel, a vaccine composed of allogeneic tumor irradiated cells combined with BCG and GMCSF as adjuvants [62]. Remarkably, in a phase III study of postsurgical adjuvant therapy involving approximately 500 stage IV melanoma patients, Canvaxin's efficacy was compared with a placebo. The results indicated that BCG/Canvaxin did not demonstrate improved outcomes over BCG/placebo [15]”.
- Figure 4 has to be reformatted. The presentation of the different plots is very confusing and difficult to read. Moreover, please add more details on the flow cytometry procedure in the material and method section.
Answer: Figure 4 has been reformatted. The updated version of Figure 4 now exclusively depicts data exhibiting statistically significant differences, the remaining information has been moved to supplementary Figure S2. A schematic summary outlining cell marker expression associated with favorable prognosis following TAPCells vaccination has been included at the bottom left of the figure. Further details on the flow cytometry procedures have been provided in the methods section.
Reviewer 3 Report
Comments and Suggestions for Authors
In their manuscript Tittarelli et al. present the results of the clinical trial, over 15 years period, that explored the potential of TAPCells, the dendritic cell vaccine, in inducing immune response for the treatment of melanoma patients. The authors analyzed the association between the clinical outcome and genetic as well as immunologic parameters of long- and short-term patient survivors. The authors report improved clinical outcome, e.g. prolonged survival, of patients that exhibit positive DTH compared to patients with negative DTH reaction. Furthermore, the authors identified specific immunomarkers associated with prolonged or reduced patient survival. To conclude, the authors emphasize the potential of vaccine-induced immune response in melanoma treatment and the relevance of identifying predictive biomarkers for TAPCells-based immunotherapy response.
The manuscript is written in a clear and understandable way, shows interesting results and I believe will contribute greatly to the field.
I only have minor remark. Due to the size, I suggest the authors to move all tables in the supplementary materials to make presentation of the results more clear to the reader.
Author Response
Answer: We sincerely appreciate the reviewer’s comments and incorporate suggested changes. As suggested, Tables 1 and 3 have been moved to supplementary material (Tables S1 and S2).
Reviewer 4 Report
Comments and Suggestions for Authors
The comments attached
